# Rounding up the Usual Suspects: Assessing Yorkie, AP-1, and Stat Coactivation in Tumorigenesis

**DOI:** 10.3390/ijms21134580

**Published:** 2020-06-27

**Authors:** Fisun Hamaratoglu, Mardelle Atkins

**Affiliations:** 1School of Biosciences, Cardiff University, Cardiff CF103AX, UK; 2Department of Biological Sciences, Sam Houston State University, Huntsville, TX 77341, USA

**Keywords:** JNK, JAK/STAT, Hippo, Notch, signaling, cancer, *Drosophila*

## Abstract

Can hyperactivation of a few key signaling effectors be the underlying reason for the majority of epithelial cancers despite different driver mutations? Here, to address this question, we use the *Drosophila* model, which allows analysis of gene expression from tumors with known initiating mutations. Furthermore, its simplified signaling pathways have numerous well characterized targets we can use as pathway readouts. In *Drosophila* tumor models, changes in the activities of three pathways, Jun N-terminal Kinase (JNK), Janus Kinase/Signal Transducer and Activator of Transcription (JAK/STAT), and Hippo, mediated by AP-1 factors, Stat92E, and Yorkie, are reported frequently. We hypothesized this may indicate that these three pathways are commonly deregulated in tumors. To assess this, we mined the available transcriptomic data and evaluated the activity levels of eight pathways in various tumor models. Indeed, at least two out of our three suspects contribute to tumor development in all *Drosophila* cancer models assessed, despite different initiating mutations or tissues of origin. Surprisingly, we found that Notch signaling is also globally activated in all models examined. We propose that these four pathways, JNK, JAK/STAT, Hippo, and Notch, are paid special attention and assayed for systematically in existing and newly developed models.

## 1. Introduction

### 1.1. Studying Tumorigenesis in Drosophila is Fast, Cheap, and Effective

*Drosophila* imaginal discs, larval precursors of adult epithelial tissues, are outstanding models of epithelial tumorigenesis. Experiments in this system have contributed enormously to our molecular and mechanistic understanding of cancer and to the development of treatments [1,2,3]. This contribution ranged from mapping downstream effectors of cancer inducing mutations to understanding synergistic interactions between driver mutations to building avatars for personalized drug screening [4,5,6]. For example, the Hippo tumor suppressor pathway was first discovered in flies [7,8,9]. The nuclear effectors of this pathway, Yorkie (Yki) in *Drosophila*, Yes Associated Protein (YAP) and Transcriptional Co-activator with a PDZ-binding Domain (TAZ) in mammals, are now established oncogenes [10,11]. Likewise, many components and signaling mechanisms of other pathways involved in cancer, such as Wnt (Wingless in *Drosophila*), Bone Morphogenetic Protein (BMP), Notch (N), Epidermal Growth Factor Receptor (EGFR)/Ras, Insulin, Janus Kinase/Signal Transducer and Activator of Transcription (JAK/STAT), and Jun N-terminal kinase (JNK), were delineated using fly genetics [12,13,14,15,16,17,18,19,20]. It is important to note that all these cancer pathways have developmental roles as well ranging from cell fate specification to regeneration. They all contribute to regulation of basic cellular processes, such as proliferation, cell growth, and apoptosis. To help the non-expert, we list the key pathway components and targets in Table 1 with a color code, where green is used for pro-growth/cancer and red for anti-growth factors.

In 2005, the Cagan Lab published a *Drosophila* model of multiple endocrine neoplasia type 2 (MEN2). They used an active form of the Ret-kinase found in patients and expressed it constitutively in larval eye discs [21]. This model was utilized to map downstream effectors and for pharmacological screening, leading to an effective, Food and Drug Administration-approved treatment for the disease within 6 years of the initial publication [22,23]. This approach has now been taken to the next level for personalized medicine, where 5–15 driver mutations from a patient are introduced into a fly avatar. These genetically personalized fruit flies are then used for drug screening [24]. This method has already proven to work and helped a patient with metastatic colorectal cancer [4]. These examples highlight the speed and power of *Drosophila* genetics for basic and translational cancer research.

### 1.2. Cooperation between Different Signaling Pathways Is a Hallmark of Tumorigenesis

Most tumors initiate as the result of multiple mutations arising, which synergistically cooperate to transform normal cells to a tumor fate, conferring proliferative potential, invasive potential, and resistance to cell death, among other characteristics [25,26]. Even in the rare cases where a single mutation has the ability to transform a cell, this is achieved by misregulation of multiple downstream signaling pathways. For example, the dominant mutations in the Ret receptor tyrosine kinase activate the Ras/ERK, Src, and JNK pathways [21]. Another example is the larval discs that are homozygous mutants for apical-basal polarity determinants *scribble (scrib)* or *discs-large (dlg)*, which form neomorphic masses [27,28]. Transformation of these mutant discs is owing to critical changes that activate JNK and Yki, leading to excess production of the JAK/STAT ligands Unpaireds (Upds) [29]. Finally, mutations in the epigenetic silencers of the Polycomb Repressive Complex 1 (PRC1) cause excess growth when the mutant clones are allowed to occupy the whole disc using the eyFLP-cell lethal system [30]. Notably, some PRC1 components have redundant functions in growth and single mutants may not always yield overgrowth [30]. Therefore, double mutant combinations were utilized; *Psc-Su(z)2* chromosome is a deletion that lacks two PRC1 components, *Posterior sex combs (Psc)* and *Suppressor of zeste 2 (Su(z)2)*, as well as an uncharacterized, non-conserved gene [30]. Similarly, the *polyhomeotic (ph)* locus is duplicated and contains proximal and distal transcription units, *ph-p* and *ph-d*, respectively [31]. The allele used to generate the *ph* tumors contains mutations in both *ph-p* and *ph-d* genes [32]. The activities of different pathways were screened in PRC1 complex mutants and the JAK/STAT pathway was found to be highly and consistently induced [30]. Accordingly, the Upd ligands were shown to be direct targets of Polycomb silencing [30]. Another study found a requirement for Notch signaling in *ph* eye disc tumors [33]. The *ph* tumors also activated the JNK pathway [34]. Therefore, deregulation of multiple signaling pathways is a common feature of tumor formation.

In 2003, ground-breaking work by Richardson and Xu labs initiated the field of cooperative tumorigenesis in *Drosophila* [35,36]. Unbiased genetic screens revealed apical-basal polarity loss as a cooperating factor with activated Ras and N mutations [35,36]. Paradoxically, if mutations in *scrib* or *dlg* genes arise in a patch of cells, the surrounding wild-type cells effectively eliminate the mutant cells, via a process known as cell competition. For recent and excellent reviews of this process, see [37,38,39,40]. However, if these *scrib* mutations in patches are combined with activated Ras or Notch, the outcome is aggressive and metastatic tumors [35,36]. With its clonal nature, this model more closely mimics the human condition. Genetics and transcriptomics revealed that JNK, JAK/STAT, and Yki are the drivers of tumorigenesis in RasV12 + *scrib-* tumors [41,42,43,44,45,46]. We set out to determine whether these three pathways, or others, are commonly deregulated in *Drosophila* tumors of different initiating mutations.

## 2. Background and Approach

### Is There a Common Signaling Signature of Tumorigenesis?

Since 2003, many labs have generated new combinations of genetic aberrations that drive tumor formation, ranging from combined defects in major signaling pathways to lysosome or mitochondrial dysfunction, or utilized the concept of cooperative oncogenesis to screen for novel synergistic interactions (for recent, comprehensive reviews, see [2,37]). Each study looked into the authors’ favorite pathways and biological processes; nevertheless, common features are evident. For example, the JNK, JAK/STAT, and Hippo signaling pathways were very frequently mentioned, leading us to question if their co-activation is truly a common feature underlying development of diverse tumors, or simply a bias in the assays used. However, as noted above, deregulation of many of the developmental signaling pathways has been implicated in tumor formation. Thus, to determine if a specific subset of pathways is consistently activated across diverse tumor types, creating a common tumorigenic signaling state, we performed a meta-analysis using available transcriptomic datasets of *Drosophila* tumors. We assessed the activity levels of JNK, JAK/STAT, Hippo, Decapentaplegic (Dpp), Hedgehog (Hh), Wingless (Wg), Notch, and EGFR/Ras pathways in tumors by surveying expression levels for selected, validated targets for each pathway. It is important to look at multiple readouts to assess pathway activity as the transcription of each gene can be and often is regulated by multiple pathways. To acknowledge this caveat, we highlighted the target genes that are known to be targets of multiple pathways in cyan in the figures. 

Notably, only a few of the *Drosophila* tumor models have been subjected to transcriptomic analysis. We compiled data from four cooperative tumorigenesis models, two single mutants that lead to transformation, and two PRC1 tumors. We also included transcriptomic data from *warts (wts)* mutants, and Notch intracellular domain (NICD) expressing discs representing hyperplastic overgrowth conditions.

We mined the transcriptomic data available for the following genotypes:(1)*dlg* mutant wing discs, data from [29]. Wing discs that are homozygous mutant for *dlg* lose polarity, leading to neoplastic overgrowth. Such discs are practically immortal and continue to grow if transplanted to new hosts [47].(2)*scrib* mutant wing discs, data from [29]. Dlg and Scrib act together in a complex and *scrib* mutant wing discs phenocopy *dlg* mutants, about 70% of all genes that are differentially expressed in *dlg* mutant wing discs are also differentially expressed in *scrib* discs; 311 upregulated and 263 downregulated genes [29].(3)Ras^V12^ + *scrib-*: This is the original and best studied cooperative tumorigenesis model. Three labs have analyzed Ras^V12^ + *scrib-* tumors from four different tissues utilizing transcriptomics [41,43,48]. Such tumors are neoplastic and metastatic [35,49].(4)N + *scrib-*: A microarray analysis was performed on N + *scrib-* tumors revealing common as well as unique changes compared with Ras^V12^ + *scrib-* tumors in the eye [43]. These tumors express NICD in *scrib**-* mitotic clones. Transcriptomes of wing discs with N + *scrib-* tumors were also published recently [50]. N + *scrib-* tumors are neoplastic in nature.(5)Abrupt + *scrib-*: The zinc finger transcription factor Abrupt was identified as a *scrib* cooperating oncogene in a screen [51]. Ectopic Abrupt expression has no discernable phenotypes on differentiation and gives the cells a slight growth advantage, whereas Abrupt overexpression in *scrib-* cells maintains cells in a progenitor-like state and prevents the formation of photoreceptors. Eye discs with such cells are severely overgrown and neoplastic [51,52].(6)*Psc-Su(z)2* double mutants, members of the Polycomb Repressive Complex 1 (PRC1). These mutations lead to dramatic overgrowth in eye and wing discs. The mutated tissues often maintain their apicobasal polarity [29,30].(7)*polyhomeotic (ph)* mutants (*ph-p* and *ph-d* double), member of the PRC1, which often display a loss of polarity along with overgrowth [33]. Clones can be invasive, and display cooperative tumorigenesis with Ras^V12^ [33]. A small proportion of animals with *ph* clones in the eye can reach adulthood and display overgrown eyes [33,53].(8)*capicua, warts (cic, wts)* double mutants: Cic is the transcriptional repressor of EGFR/Ras signaling [54]. The Ras/Raf/MAPK Kinase (MEK) cascade culminates in activation of Mitogen-Activated Protein Kinase (MAPK), which targets Cic for degradation, allowing target gene induction [55]. Wts kinase acts in the Hippo pathway [49,56,57,58,59,60]. In *wts* mutants, Yki is stabilized and can accumulate in the nucleus and help induce expression of genes that drive cell growth (such as Myc), proliferation (e.g., Cyclin E), as well as conferring apoptotic resistance via induction of Diap1 [7,8,61]. Hippo signaling controls the transcriptional output of the Ras pathway and their mutual disruption, as in *cic,wts* mutants, causes synergistic overgrowth in larval discs [54]. Such discs stay hyperplastic and lose apical-basal polarity only at the very late stages [54]. We have data for day 5 (prior to overgrowth) and with overgrown, heavily folded day 9 discs [54].(9)*wts* mutants at day 5 and day 9, data from [54]. These discs are hyperplastic owing to overactivation of Yki.(10)NICD-overexpressing wing discs display hyperplastic overgrowth, data from [50].

We exclusively used datasets generated from imaginal disc tissue where the fold change in tumor versus a wild-type control was available. The only exception to this rule is the *ph* tumor dataset; in this study, the tissue was subjected to Fluorescence-Activated Cell Sorting (FACS) prior to RNA-sequencing and the fold changes in tumor cells labeled with Green Fluorescent Protein (GFP) were reported against the neighboring GFP− cells [53]. We reported expression values in the figures below as fold change in log2 (log2FC) compared with controls. As we are using these values as proxies for pathway activity, we are reporting all values that are differentially expressed (*p* < 0.05) without a fold change (FC) threshold. The published *ph* dataset used a very stringent cut-off (padj < 0.01), and thus values for many genes were not available, shown in dark grey in the figures. Finally, the various datasets are annotated to different releases of the *Drosophila* genome. When genes of interest were not found, attempts were made to query all known synonyms, but we cannot preclude that some genes may have been missed owing to the annotation differences.

## 3. Analysis/Results

### 3.1. Tumors Cause Delayed Pupariation and Loss of Cell Fate Specification 

Notably, a phenotype shared between all the conditions assessed in this analysis is the heavily delayed pupariation, which manifests as giant larvae. This phenotype is known to be caused by excess *Drosophila* insulin-like peptide 8 (Dilp8) production [62]. Dilp8 secreted from discs acts remotely on the central brain to delay the transition to the pupal stage. As such, larval life is extended, in some cases indefinitely, giving the tumors more time to grow [63,64]. Among the conditions we re-analyzed, *dilp8* induction was consistently very strong, with the exception of the *ph* tumors, ranging from 3.23-fold in Ras^V12^ + *scrib-* leg discs to an over 1000-fold induction in Ras^V12^ + *scrib-* wing discs (Figure 1). *dilp8* is a verified Yki/Sd target gene, but it can also be induced by JNK or JAK/STAT activation [63,64,65]. Notably, in cases where Yki is not activated, such as in *Psc-Su(z)2* mutants, *dilp8* is still heavily induced, showing that it can be induced independently of Yki activation (Figure 1).

Another common feature across various tumors is the loss of cell fate specification. In eye tumors, defects in photoreceptor differentiation can be easily revealed by Embryonic Lethal Abnormal Vision (ELAV) staining; such defects were reported for Ras^V12^ + *scrib-*, Ab + *scrib-, cic,wts,* and *ph* tumors [45,51,53,54]. We analyzed the expressions of the fate determinants for the eye (*twin of eyeless,*
*toy; eyeless, ey: eyes absent, eya; sine oculis, so; dachshund, dac; ELAV; atonal, ato; homothorax, hth; teashirt, tsh; senseless, sens;*) and the wing tissue (*hth; tsh; sens; Distalless, Dll; vestigial, vg; nubbin, nub*). Neoplastic tumors and PRC1 tumors showed significant downregulation of corresponding fate determinants (Figure 1, yellow and blue boxes). These changes were less pronounced in *wts* mutant and NICD overexpressing wing discs, as well as in *cic,wts* mutant discs (Figure 1). The antenna and the leg tumors showed a signature similar to that of the eye tumors. Thus, we observe that not only is differentiation lost (*ato*, *sens*, *ELAV*), but much of the early tissue identity program is also downregulated during or as a result of neoplastic tumor formation in both wing and eye discs (Figure 1). We then investigated the activity of different signaling pathways to assess if common processes are activated to block development and promote tumorigenesis.

### 3.2. Inactivation of Hippo, Dpp, Hh, Wg, and Activation of JNK, JAK/STAT, and Notch Are Commonly Seen in Tumors

**EGFR/Ras pathway:** Notably, two of the four cooperative tumorigenesis models are based on activation of the EGFR/Ras pathway, and as expected, transcriptional feedback regulators of this pathway Argos (Aos), Sprouty (Sty), and Sulfated (Sulf1) were significantly induced in Ras^V12^ + *scrib-* and *cic,wts* tumors (Figure 2). In addition to the negative feedback regulators of the pathway and its transcription factor Pointed (Pnt), all of which are direct Cic targets, we included the newly identified Cic target genes *Ecdysone receptor (EcR*), EGF *spitz (spi),* and *Leucine-rich tendon-specific protein* (*Lrt*) in our analysis [54]. EGFR/Ras activity was largely unaffected in the Abrupt + *scrib-*, *scrib*, and *dlg* tumors. The pathway activity was lowered in the N + *scrib-* and *ph* eye tumors. Thus, EGFR pathway activation is a specific feature of a subset of tumors.

**Dpp, Hh, and Wg pathways**: The activities of each of these developmental pathways were very low in all tumor models examined, with a few notable exceptions. The shutdown of these pathways is likely to reflect defects in differentiation, known to occur in these tumors [51,53,54] (Figure 1). 

For the Dpp pathway, all the target genes displayed are induced upon Dpp activation with the exception of *brinker (brk)*, which is shut down by Dpp signaling [66,67]. Therefore, induction of *brk* transcription indicates low Dpp activity. On the basis of upregulation of this gene, Dpp signaling is markedly reduced in *Ras^V12^ + scrib-* and *cic,wts* tumors (Figure 3). Curiously, Dpp signaling is high in *ph* eye disc tumors (Figure 3). Strikingly, we noticed that the secreted BMP antagonist *short gastrulation (sog)* [68,69] was upregulated in the *Ras^V12^ + scrib-* transcriptomes. We queried this across the tumor transcriptomes and found a strong correlation between *sog* expression and Dpp pathway activity. We found that *sog* is upregulated in all tumors assayed, except *ph,* where it is strongly downregulated (log2FC = −4.16). Therefore, *sog* upregulation offers a potential mechanistic explanation for the observed downregulation of Dpp activity levels. Interestingly, *sog* transcript levels are not changed and the Dpp activity is low in the *Psc/Su(z)2* wing tumors; this difference to *ph* tumors may be owing to different tissue types.

For the Hedgehog pathway, most pathway members/targets assessed were lower or unchanged in tumors, suggesting there may be little flux through this pathway in tumors. A notable exception is the upregulation of *dpp* ligand expression in *scrib*, *dlg*, and *PRC1* tumors. It is not clear at this level of analysis if *dpp* expression in these cells is dependent on Hh signaling or induced by a different factor.

Similarly, the activity of the Wg pathway is downregulated in most of the tumors assayed except the PRC1 tumors and NICD-expressing discs. Wg expression is known to be regulated by N signaling [70,71]. In the PRC1 tumors, Wg signaling is likely induced in the presence of excess ligand production (Wg, Wnt4, and Wnt6), highlighting another difference between PRC1 tumors and the others we analyzed.

Our analysis supports the idea that, in most neoplastic tumors, normal developmental signaling is not functioning. However, it highlights that the PRC1 tumors may be largely different from the more neoplastic tumors assessed. Interestingly, Torres et al. performed a hierarchical clustering between *ph* tumor transcriptomes and other known normal and tumor transcriptomes. This analysis demonstrated that the *ph* tumors were more similar to embryonic signatures rather than other tumors (e.g., Ras^V12^ + *scrib-*), which were very dissimilar from embryonic signatures [53].

**Hippo pathway**: Despite the impressive overgrowth phenotypes of *hippo* mutant imaginal discs, the involvement of the Hippo signaling pathway is not assayed for consistently in tumor models. Only one out of every four papers that deal with tumor models reported on Hippo activity levels. The first member of the Hippo cascade, Warts (Wts) kinase, was described as a tumor suppressor in flies in the 1990s [72,73]. Wts stayed as an orphan kinase until the discovery of Salvador and Hippo in the early 2000s [49,56,57,58,59,60,74]. Since then, the Hippo signaling cascade has well reached adulthood and has proven itself as a key regulator of cell division and stemness in many contexts [9,10,11,75,76]. Yki is the main transcriptional mediator of this tumor suppressor pathway. Inactivation of the Hippo pathway leads to activation of Yki and subsequent tissue growth [77]. Yki localization is linked to pathway activity. Yki can accumulate in the nucleus only after the Hippo pathway is shut down [77]. Therefore, subcellular Yki localization and sensitivity to *yki* gene dosage are good assays for testing involvement of the Hippo pathway. It is trickier to interpret experiments where *yki*-RNAi is shown to suppress tumor growth as *yki*-RNAi prevents proliferation of normal cells as well. In addition to Yki localization, other good readouts for Yki activity in imaginal discs are *expanded(ex)-lacZ*, *Diap1-lacZ*, and the bantam sensor [57,78,79,80]. In RNA-seq experiments, the target genes *ex, kibra*, and *dilp8* are particularly robust readouts, whereas *CycE, Myc,* and *Diap1* are upregulated more mildly, about 1.5-fold in *wts* mutant discs (Figure 4 and Figure 9). Notably, high levels of F-actin are a well characterized and potent activator of Yki [81,82]. Therefore, tumors where actin accumulation was observed may have higher Yki activity [51,52,83,84,85,86].

In our meta-analysis of sequenced tumors, Yki target genes were expressed in all tumors at levels similar to those observed in *wts* mutants, except for the PRC1 tumors. As *wts* loss leads to constitutive Yki activation, this should represent a strong induction of validated targets. In Ras^V12^ + *scrib-* tumors, Yki activation was less obvious in eye disc tumors compared with other tissues (Figure 4). Thus, Yki activation is a broad, but not ubiquitous phenomenon in tumors. In light of this, it is possible to speculate that Yki activation status in a tumor may be a potent way to stratify tumors that activate distinct sets of pathways to promote their growth. Tumors that do not activate Yki may become neoplastic via a different mechanism and may behave quite differently from those that do activate Yki.

**JNK pathway**: The JNK pathway is stimulated by the Tumor Necrosis Factor (TNF)-type ligand Eiger binding to its receptors Wengen and Grindelwald, which activate a MAP kinase cascade that ultimately phosphorylates the downstream Jun-kinase Basket (Bsk) [87,88,89,90,91,92]. Bsk in turn activates numerous AP-1 and AP-1 related b-Zip transcription factors. In *Drosophila* tumors, the b-Zip transcription factors Jra, Kay, Atf-3, Pdp1, and Irbp18 have been implicated as potential effectors [41,43,44,50,93,94].

JNK pathway activity is well monitored in tumors; 30 out of 40 recent papers we analyzed across various tumor models directly assayed for JNK activity in their models and with good reason. JNK signaling is activated in response to stress and tumor formation is no doubt a source of stress for the tumor initiating cell and its neighbors [17,18]. Whereas JNK activation can be anticipated in the presence of tumors, the outcome of JNK activation is not easily predicted. One of the JNK target genes, *Matrix metalloproteinase 1 (Mmp1)*, can lead to basement membrane degradation, and hence is associated with invasive behavior [44]. Moreover, JNK activation in a cell can induce autonomous and cell non-autonomous proliferation via induction of the Upd ligands [19]. Finally, it can also induce autonomous cell death via induction of the pro-apoptotic gene *reaper (rpr)* [19].

In our analysis of the literature, we observed that the outcome of JNK signaling in tumors can be as diverse as cellular responses to it. For comprehensive reviews of pro- and anti-tumorigenic roles of JNK signaling, see [19] and [95]; however, we will briefly summarize some highlights here. JNK induction is commonly a contributing factor to tumor growth and metastasis, as co-expression of a dominant-negative version of JNK (Bsk-DN) can suppress tumor growth and invasion in Ras^V12^ + *scrib-* and N + *scrib-* tumors [44,46,48]. In *ph* tumors, co-expression of Bsk-DN reduces the volume of mutant clones [34]. Similarly, JNK activity is induced and co-expression of Bsk-DN can suppress tumorigenesis in Ras + Pico [96], Ras + Src [83,97], and activated N + Src [98] models. However, in some tumors, tumor growth and invasive potential can be uncoupled from each other. In Ab + *scrib-* tumors, co-expression of Bsk-DN suppresses the invasive behavior, but not the tumor growth [51]. In contrast, the tumors are even larger owing to an extended larval life, but have a more hyperplastic appearance [51]. Therefore, it is possible to achieve excessive overgrowth independently of JNK, however, invasive behavior is tightly linked to high JNK activity.

Finally, we came across two other studies where JNK activation may have a tumor suppressive quality because co-expression of Bsk-DN actually enhances the overgrowth observed; that is, *eyeful* tumors [99] and the polyploid tumors formed by cooperation between cytokinesis failure and activated Ras or Yki expression [100]. Therefore, the outcome of JNK activation is very much context-dependent. Many other studies reported JNK induction in their respective tumor models, but did not assess the nature of JNK contribution.

The JNK pathway was globally induced in all tumors and hyperplastic discs that were subjected to transcriptomics (Figure 5). This induction is rather strong and clear in all tumors with the notable exception of the *cic,wts* tumors at day 5 (Figure 5). At this stage, *cic,wts* discs are only slightly overgrown and retain their apical-basal polarity. *Mmp1* and *upd1–3* ligands are further induced at day 9 *cic,wts* tumors, but this induction is still relatively mild compared with other tumors and more similar to JNK activity levels in *wts* mutant discs. Therefore, JNK activity seems to increase over time in the *cic, wts* model. The same trend was also observed in the Ras^V12^ + *scrib-* eye tumors [42]. As more single cell sequencing data emerges, it will be interesting to see if this trend is biologically relevant, or if it is simply a reflection of the shifted ratio of tumor to normal cells in older tumor samples in bulk RNA-seq experiments. If it is biologically relevant, it will be interesting to find if this is a general trend for all tumors and whether there is a threshold of JNK activity where invasive behavior can be initiated. We observe that JNK activation is a common feature, but with conflicting outcomes. Thus, we advocate that pathway activity be assessed, but also blocked in tumors going forward to assess if JNK has pro-tumorigenic or tumor suppressing functions in each context. Increased data points may paint a clearer picture of how the tumor context or progression correlates with JNK function.

**JAK/STAT pathway:** The workings of the JAK/STAT pathway are relatively simple. The interleukin (IL)6 type ligands Upd1 (that is, os), Upd2, and Upd3 bind to their receptor Domeless, which stimulates its dimerization and activation of the associated Janus Kinase (Jak) molecule Hopscotch (Hop). Hop, in turn, phosphorylates the sole Stat ortholog in *Drosophila*, Stat92E [20]. Activation of Stat92E promotes the growth, proliferation, invasion, and survival of tumor cells [41,42,45,101]. The ligands *upd1, 2,* and *3* are often upregulated in tumors, whereas the other pathway components are not differentially expressed [29,41].

JAK/STAT activity can be assessed by expression of its best studied target gene, *Suppressor of Cytokine Signaling at 36E (Socs36E)* in various contexts [42,65,102,103,104]. Socs36E acts as a feedback inhibitor of the pathway and can be used to test the contribution of JAK/STAT activity to tumors. Co-expression of *UAS-Socs36E* can suppress PRC1 tumors [30], overgrowth of *dlg* homozygous mutant discs [29], and the Hippo/Ras synergy [54]. Likewise, Davie et al. demonstrated that null mutations in *Stat92E* strongly suppressed the excessive growth of Ras^V12^ + *scrib-* eye tumors [42]. Finally, introducing heterozygosity for *Stat92E* into the tumor background reduced the size of *Psc-Su(z)2* eye tumors [30]. Thus, in all reported cases, excess JAK/STAT activity contributes to tumor growth. Ptp61F, a protein tyrosine phosphatase, is another negative regulator and transcriptional target of the pathway [42,105,106]. Interestingly, *Socs36E* and *Ptp61F* are already broadly upregulated in tumors, leading us to speculate that the aggressive phenotypes observed would be more severe if these genes were mutated or downregulated. 

Several other direct targets of the pathway have been identified and verified by the Bach Lab over the years, including *chronologically inappropriate morphogenesis (chinmo)*, *cap-n-collar* (*cnc*), *lamina ancestor (lama), Zn finger homeodomain 2* (*zfh2*), *NADPH oxidase* (*Nox*), *Keap1 I* (*Keap1*), and *pannier* (*pnr*) [65,101,107,108]. Among these, *Nox*, *Keap1*, and *pnr* are negatively regulated by the pathway and thus are shown in bold and traced in red (Figure 6). CG1572 was identified as an effector of the pathway in hematopoietic tumors [104]. *terribly reduced optic lobes* (*trol*), which encodes a Perlecan, is regulated by the pathway in multiple tissues [42,107]. Finally, an embryonic target gene *dorsal* (*dl*) was identified as a likely direct Stat92E target gene and had more accessible chromatin in Ras^V12^ + *scrib*- eye tumors and Upd overexpressing eye discs [42,109]. In agreement with the published results, these selected target genes indicated increased JAK/STAT activity in nearly all tumor models examined as well as the hyperplastic *wts* mutant, and NICD-expressing discs (Figure 6). In the case of *scrib*, *Ab* tumors, about half of the target genes indicate pathway activation, whereas the rest strongly argue the opposite, preventing us from reaching a verdict.

Our analysis was also inconclusive in *ph* tumors despite higher levels of Upds, but these tumors are known to have high JAK/STAT activity [30,34]. Previous studies have shown that tumor produced Upds stimulate JAK/STAT signaling in the neighboring cells [19,45,110]. In the *ph* dataset, gene expression in tumor cells was compared to that of non-tumor neighbors [53], which likely also have high JAK/STAT activity. Thus, this approach may have filtered out the differential expression of Stat target genes. Upds are transcriptional targets of JNK signaling, thus JNK activation should cause activation of JAK/STAT signaling. Indeed, this seems to be the case, with the exception of the Ab + *scrib-* model, where many JNK targets are induced, but not the Upds (Figure 5). This observation was rather surprising considering that Yki and JNK are both active in this tumor, and Upd3 is a known cooperative target of these pathways [29,51]. However, as noted above, JNK’s function in Ab + *scrib-* tumors is pro-invasive, but not pro-growth. As JAK/STAT signaling is a potent pro-growth pathway, it is possible that this model, in which we observe an uncoupling of JNK + Yki from JAK/STAT activation, could be an important system in which to gain further insights into cross talk between these pathways in tumors. It would be interesting to examine if the *upd* promoter regions are accessible in these tumors. Therefore, why the JAK/STAT pathway is not activated in this tumor is quite interesting, but the mechanism remains unknown.

**Notch pathway:** Activation of Dpp, Hh, and Wg pathways as well as EGFR or N signaling can induce cell proliferation and hyperplastic tissue growth, apart from and in addition to their roles in differentiation. The amount of growth that can be induced is especially impressive upon N activation [71]. However, what regulates the switch between differentiation and proliferation control is often poorly understood. In the case of Ras signaling, co-activation of Yki was proposed to shift the balance towards proliferation [54]. Pathways involved in cell fate specification in imaginal discs, Dpp, Hh, and Wg showed overall lower activity in tumors (Figure 3). On the contrary and to our surprise, we find that N signaling seems to be activated in all tumors examined (Figure 7). This is less clear from our target analysis in the case of the *ph* tumors (Figure 7), however, Notch activation was experimentally validated in these tumors [33,34].

In *Drosophila,* Notch activation has been identified as being able to cooperate with both the loss of *scribble* or the activation of Src to promote tumor development [48,50,98]. Src activation synergizes with N in inducing hyperplastic growth in eye and wing discs, and leads to JNK activation [98]. In combination with *scrib* loss, N activation leads to neoplastic and invasive tumors [48]. bHLH transcription factors *deadpan (dpn)* and the *Enhancer of Split (E(spl))* genes are the best characterized and direct target genes for N during differentiation, however, these canonical N targets are sometimes downregulated in N induced tumors [71,98]. Indeed, in N + *scrib-* tumors, only 3 out of 13 of these bHLH factors were induced (Figure 7). Expression of *E(spl)* genes was broadly downregulated in diverse tumor types (Figure 7). Ho et al. reported that Src can inhibit transcription of *E(spl)* genes [98]. Thus, we also checked Src levels, but overall found them to be lower in tumors (Figure 7). Therefore, reduced *E(spl)* transcription is not owing to higher Src levels in these tumors.

N signaling is peculiar as the receptor N is also the transcription factor of the pathway. Upon ligand binding, N undergoes a series of cleavages and its intracellular domain, NICD, translocates into the nucleus. N forms a complex with Suppressor of Hairless (Su(H)) and Mastermind, to induce transcription of Notch-responsive genes [14,111]. The Bray lab mapped Su(H) binding sites genome wide and asked which genes were induced upon N activation in *Drosophila* cells and in wing disc tumors, defining additional direct N target genes [71,112]. Instead of the canonical N target genes, a whole set of other genes was induced during N hyperplasia; 58 genes were commonly induced in NICD and Su(H) expressing overgrown discs and had Su(H) binding in their vicinity [71]. Of these 58, 9 representative target genes (*Serrate, Ser; Kank; zormin; fruitless, fru; CG6191; CG3835; CG18507; pickled eggs, pigs*; and *jitterbug, jbug*) are shown in Figure 7. The rate-limiting enzyme in glycolysis, Lactate dehydrogenase encoded by *ImpL3/Ldh*, was also shown to be a direct N target placing glucose metabolism downstream of N activity, and hence was included in our analysis [113]). These ten readouts were broadly activated in all the tumors assessed as well as in *wts* mutant discs and NICD-expressing discs (Figure 7). We noted that about 20% of N targets in tumors are also high confidence Yki/Sd targets [41,71]. These included target genes that are regulated by both the Hippo and JNK pathways: *chinmo*, *ftz-f1*, *upd2*, and *upd3* loci. These loci are some of the most highly and consistently upregulated genes across tumors we assessed, raising the possibility that Notch activation also contributes to promoting the robust upregulation of these important genes. Clearly, these results indicate that there remains much to be discovered about the role of Notch signaling in these tumors, and its possible relationships with AP-1, Stat, and Yorkie dependent gene expression.

### 3.3. A Set of Transcription Factors Is Commonly Upregulated in Tumors

From the available transcriptomics studies, several transcription factors have also been identified as key regulators of tumor progression/gene expression primarily downstream of these pathways. We decided to also query these transcription factors to see if their upregulation was a common feature. Thus, we examined the expression of selected AP-1 transcription factors (Kay, Atf3, Irbp18, Jra, Pdp1), Myc, Ftz-f1, Ets-21C, Chinmo, Abrupt, and the repressor Cic (Figure 8). Interestingly, *cic* was downregulated in most tumors to a similar degree as it was by mutation in the *cic,wts* model. *ftz-f1*, an ortholog of the human orphan nuclear receptor 5A (NR5A), was mildly to strongly upregulated in nearly every model. *Atf3* and *Pdp1* were also notably upregulated as well as *Ets21C* and *chinmo*. The broad upregulation of these factors may suggest that they bear further investigation in other emerging models as each has orthologs that have been implicated in human tumorigenesis.

Finally, we also analyzed the expression of apoptosis and cell cycle regulators in the tumor models. In agreement with the literature, *wts* and *cic,wts* mutant discs had higher levels of the anti-apoptotic gene *Diap1* and lower levels of the pro-apoptotic gene *reaper (rpr*), protecting them from cell death (Figure 9) [8,114]. Furthermore, CycE and String (Stg), rate limiting factors in cell cycle, were induced in *wts* and *cic,wts* discs, leading to excess cell divisions (Figure 9) [9]. NICD expressing discs behaved similarly (Figure 9). The other tumors seemed to have overall higher expression of the pro-apoptotic genes and reduced levels of the cell cycle genes, which appeared counterintuitive given their proliferating state (Figure 9). High *rpr* expression can be attributed to JNK activity, however, regulation of *grim* in Ras^V12^ + *scrib-* tumors begs for further attention, as it was consistently induced in all four tissues (Figure 9). The upregulation of these regulators of apoptosis may be relevant to the “undead” phenomenon of sublethal caspase activation reported in Ras-dependent tumors [97,115]. Alternatively, this may reflect cell death induction in either the tumor or neighboring cells. Future single-cell experiments may be able to better differentiate the role of these genes and the apoptotic pathways in the growth or restraint of the tumor, as this analysis cannot differentiate autonomous versus non-autonomous gene induction and is limited by the single time point.

## 4. Discussion and Conclusions

In the quest to develop better therapeutic approaches, there has been a push towards personalized medicine based on patients’ mutational status. The use of sophisticated statistical and computational analyses of genomic data from human tumors revealed large numbers of candidate genes and variants. While this has yielded some clear gains, it is often tough to interpret this big cancer data and to translate it to improved patient care [116]. The way forward will be paved by functional studies in experimental models to reveal the biological significance of the newly identified variants [117]. Here, we sought to determine if tumors with different known initiating mutations share common cell biology in the form of signal pathway activity. If so, this may indicate that unique mutational signatures converge on common signaling states to create tumors. By identifying common core signaling states, tumors may be stratified for potential treatment with novel inhibitor combinations. Owing to its demonstrated successes as an initial discovery platform as well as a translational model for tumor therapies, insights from *Drosophila* will continue to be informative in the age of personalized cancer medicine.

We observe in our meta-analysis that, while not a rule, Hippo pathway is inactivated, whereas Notch, JNK, and STAT signaling pathways are frequently co-activated in *Drosophila* tumors (Figure 10). It is already known that co-activation of Yki and JNK can synergistically activate *upd* expression, leading to Stat activation [29]. It would be interesting to know if there is additional cross-regulation between the pathways, which could lead to a self-sustaining circuit in tumors that makes them independent of initiating mutations once the circuit of JNK/Yki/Stat is established.

We were surprised to discover that the data we analyzed supported that the Notch pathway is also commonly activated in different models. To our knowledge, it is not known if Notch signaling is important for the growth or invasive properties of most of the tumors assessed in this study, or how it would become activated in, for example, Ras^V12^ + *scrib-* or *cic,wts* tumors. Furthermore, interaction of N with Yki or Stat in tumors has been little explored, potentially opening new avenues of investigation and insights into the altered signaling milieu of tumors. This finding highlights the strength and merit of these kinds of meta-analyses. 

While activation of Yki, JAK/STAT, and JNK is commonly observed in tumors, deregulation of all three simultaneously is not an obligate scenario. As noted, PRC1 tumors did not activate Yki, whereas upregulation of Upds and JAK/STAT induction is uniquely missing in Ab + *scrib-* tumors. Our analysis suggests that the *ph* tumors activate Dpp and Wg pathways, unique among the tumors assessed. This tumor also showed a signature that was quite similar to normal embryonic gene expression, representing a very different kind of tumor [53]. Overall, with the frequent exception of the *ph* tumors, the key factors induced are *chinmo, Ets21C, ftz-f1, Atf3*, and *Pdp1*, forming a common tumor signature found across tumors with different initiating mutations (Figure 8). Further documentation of which tumors do or do not rely on the Yki/Stat/AP-1 axis could yield a new broader way to stratify tumors. In turn, it is possible this insight could lead to simplified or more effective treatment strategies.

## Figures and Tables

**Figure 1 ijms-21-04580-f001:**
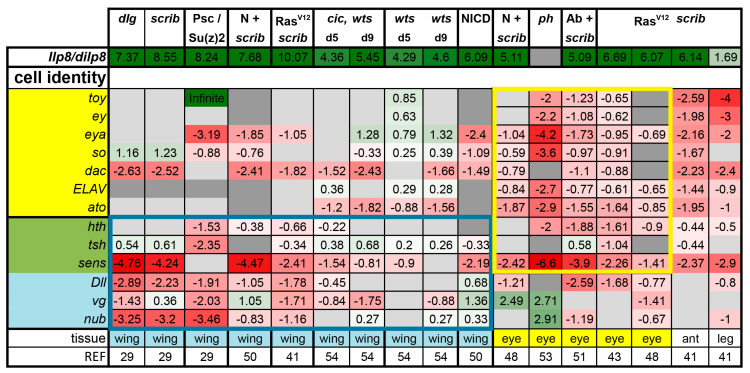
Expression levels of *dilp8* and cell fate determinants in various tumor models. Samples were grouped according to their tissue of origin. Genes that act in the retinal determination network and their downstream effectors are highlighted in yellow, whereas genes required for cell fate determination in the wing are highlighted in blue. Three genes with developmental roles in both tissues are highlighted in green. Gene expression is displayed as fold change in log2 (log2FC) compared with controls. Color coding corresponds to a heatmap with a scale from −5 (dark red, downregulation) to 5 (dark green, upregulation). The midpoint, 0, is white. Light grey cells represent non-significant changes with *p* ≥ 0.05. Cells are shown in dark grey if the gene was not found in the datasets owing to different cut-offs used by different groups. Abbreviations: Insulin like peptide 8, Ilp8; *twin of eyeless,*
*toy; eyeless, ey; eyes absent, eya; sine oculis, so; dachshund, dac; Embryonic Lethal Abnormal Vision, ELAV; atonal, ato; homothorax, hth*; *teashirt, tsh; senseless, sens; Distalless, Dll; vestigial, vg; nubbin, nub*).

**Figure 2 ijms-21-04580-f002:**
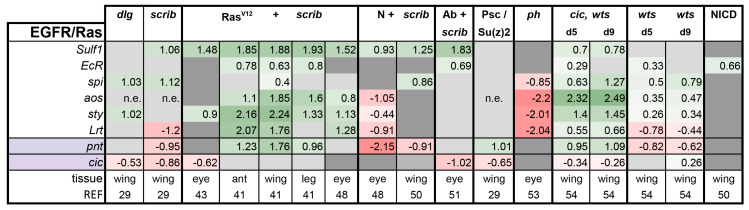
Epidermal growth factor receptor (EGFR)/Ras activation is not a prerequisite for tumor formation. Light purple highlights the transcription factors of the pathway. Note that *pnt* is a transcription factor and also a transcriptional target of the pathway. Gene expression is displayed in log2FC. Color coding corresponds to a heatmap with a scale from −5 (dark red, downregulation) to 5 (dark green, upregulation). Light grey cells represent non-significant changes with *p* ≥ 0.05. Dark grey: the gene was not found in the dataset. *n*.e. = not expressed, the gene was in the data, but showed 0 reads. Abbreviations: *Sulfated, Sulf1; Ecdysone receptor, EcR spitz, spi; argos, aos; sprouty, sty; Leucine-rich tendon-specific protein*, *Lrt*; *pointed, pnt*.

**Figure 3 ijms-21-04580-f003:**
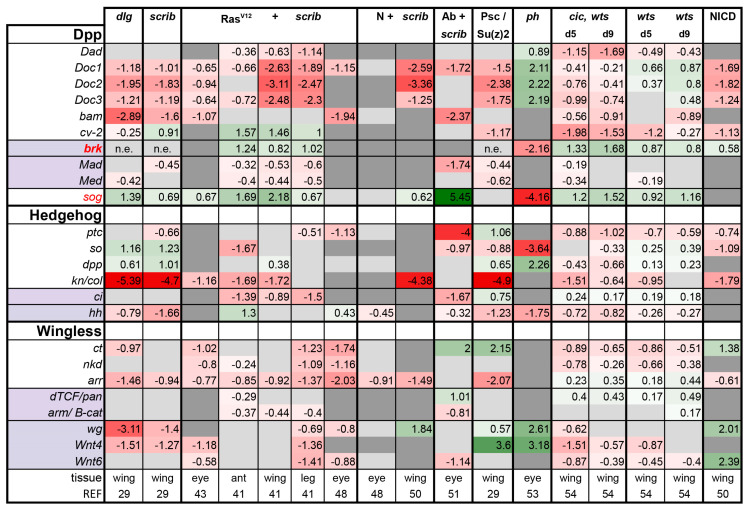
Tumors display a trend of overall suppression of Dpp, Hh, and Wg pathways. The target genes that are suppressed by the pathway are shown in bold, and pathway inhibitors are shown in red. Light purple highlights the transcription factors of each pathway. *brk* meets all three criteria. In the case of the Hh and Wg pathways, the ligands are also shown and highlighted in light purple, their expression is not necessarily regulated by their own pathway, but their excess production can activate their respective pathway. *dpp* is a transcriptional target of Hh signaling and is shown in that section. Gene expression is displayed in log2FC. Color coding corresponds to a heatmap with a scale from −5 (dark red, downregulation) to 5 (dark green, upregulation). Light grey cells represent non-significant changes with *p* ≥ 0.05. Dark grey: the gene was not found in the dataset. *n*.e. = not expressed, the gene was in the data, but showed 0 reads. Abbreviations: *Daughters against Dpp, Dad; Dorsocross 1-3, Doc1-3; bag of marbles, bam; crossveinless-2, cv-2; brinker, brk; Mothers against Dpp, Mad; Medea, Med; short gastrulation, sog; knot, kn; collier, coll; cut, ct; naked, nkd; arrow, arr; Drosophila T cell factor, dTCF; pangolin, pan; armadillo, arm.*

**Figure 4 ijms-21-04580-f004:**
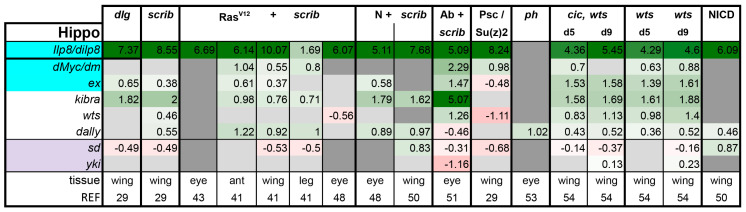
Yorkie activation contributes to tumorigenesis in models with polarity loss. Light purple highlights the nuclear factors of the pathway. Target genes that are known to also be regulated by other pathways are highlighted in cyan. Gene expression is displayed in log2FC. Color coding corresponds to a heatmap with a scale from −5 (dark red) to 5 (dark green). Light grey cells represent non-significant changes with *p* ≥ 0.05. Dark grey: the gene was not found in the dataset. Abbreviations: *Insulin like peptide 8, Ilp8; diminutive, dm; expanded, ex; warts, wts; division abnormally delayed, dally; scalloped, sd; yorkie, yki.*

**Figure 5 ijms-21-04580-f005:**
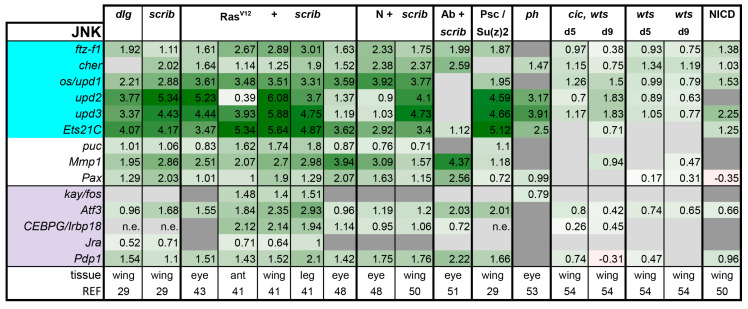
Activation of JNK is a common feature of all tumor models analyzed. Light purple highlights nuclear factors of the JNK pathway. Target genes that are known to also be regulated by other pathways are highlighted in cyan. Gene expression is displayed in log2FC. Color coding corresponds to a heatmap with a scale from −5 (dark red) to 5 (dark green). Light grey cells represent non-significant changes with *p* ≥ 0.05. Dark grey: the gene was not found in the dataset. *n*.e. = not expressed, the gene was in the data, but showed 0 reads. Abbreviations: *ftz transcription factor 1, ftz-f1*; *cheerio, cher*; *outstretched, os*; *Ets at 21C, Ets21c*; *Paxillin, Pax; kayak, kay; Activating transcription factor 3, Atf3; CCAATT Enhancer Binding Protein Gamma, CEBPG; Inverted repeat binding protein 18, Irbp18; Jun-related antigen, Jra; PAR-domain protein 1, Pdp1.*

**Figure 6 ijms-21-04580-f006:**
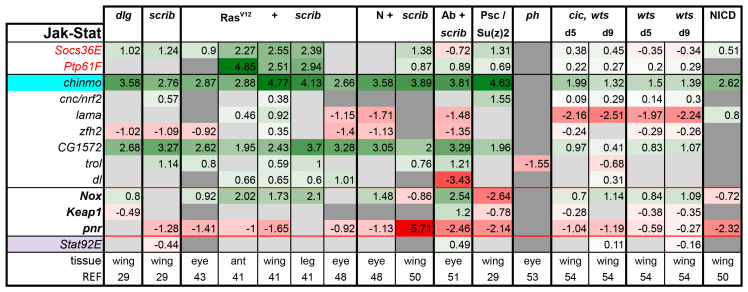
Most tumors have high levels of JAK/STAT activity. The target genes that are suppressed by the pathway are shown in bold, and pathway inhibitors are shown in red. Light purple highlights the transcription factor of the pathway. Gene expression is displayed in log2FC. Color coding corresponds to a heatmap with a scale from −5 (dark red) to 5 (dark green). Light grey cells represent non-significant changes with *p* ≥ 0.05. Dark grey: the gene was not found in the dataset. Abbreviations: *Suppressor of Cytokine Signaling at 36E, Socs36E*; *Protein tyrosine phosphatase 61F, Ptp61F; chronologically inappropriate morphogenesis, chinmo; cap-n-collar,*
*cnc; lamina ancestor, lama; Zn finger homeodomain 2, zfh2*; *terribly reduced optic lobes,*
*trol; dorsal, dl; NADPH oxidase, Nox*; *Keap1, Keap1*; and *pannier, pnr*.

**Figure 7 ijms-21-04580-f007:**
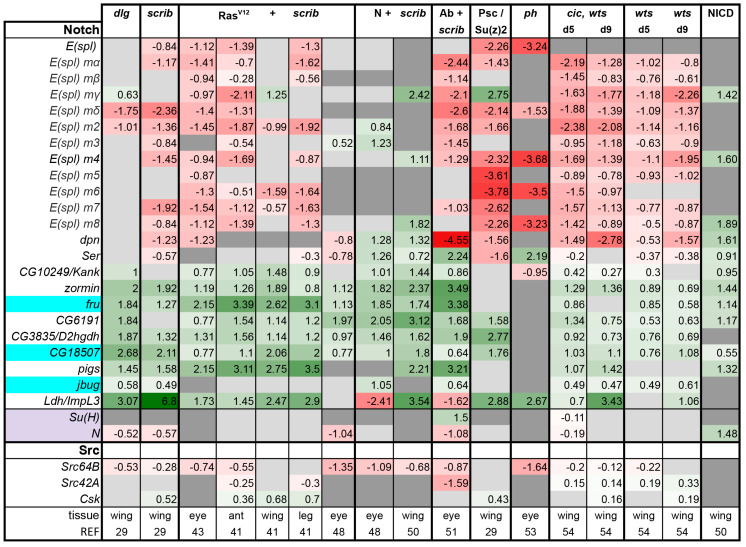
E (spl) genes are downregulated, whereas the other N target genes are induced in tumors. Light purple highlights the nuclear factors the pathway. Genes highlighted in cyan are predicted Yki/Sd targets [41]. Gene expression is displayed in log2FC. Color coding corresponds to a heatmap with a scale from −5 (dark red) to 5 (dark green). Light grey cells represent non-significant changes with *p* > 0.05. Dark grey: the gene was not found in the datasets. Abbreviations: Enhancer of split, E(spl); deadpan, dpn; Serrate, Ser; fruitless, fru; D-2-hydroxyglutaric acid dehydrogenase, D2hgdh; pickled eggs, pigs; jitterbug, jbug; Lactate dehydrogenase, Ldh; C-terminal Src kinase, Csk.

**Figure 8 ijms-21-04580-f008:**
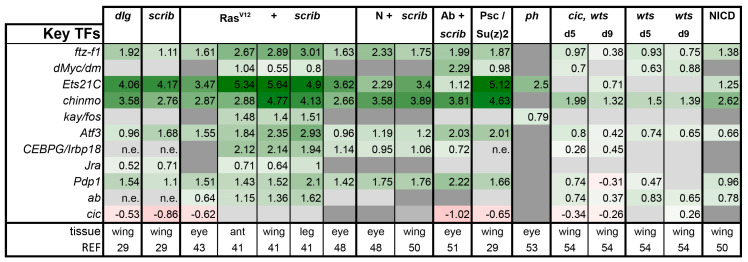
Expression of select transcription factors (TFs) across tumors. Gene expression is displayed in log2FC. Color coding corresponds to a heatmap with a scale from −5 (dark red) to 5 (dark green). Light grey cells represent non-significant changes with *p* ≥ 0.05. Dark grey: the gene was not found in the datasets. *n*.e. = not expressed, the gene was in the data, but showed 0 reads. Abbreviations: ftz transcription factor 1, ftz-f1; diminutive, dm; Ets at 21C, Ets21c; chronologically inappropriate morphogenesis, chinmo; kayak, kay; Activating transcription factor 3, Atf3; CCAATT Enhancer Binding Protein Gamma, CEBPG; Inverted repeat binding protein 18, Irbp18; Jun-related antigen, Jra; PAR-domain protein 1, Pdp1; abrupt, ab; capicua, cic.

**Figure 9 ijms-21-04580-f009:**
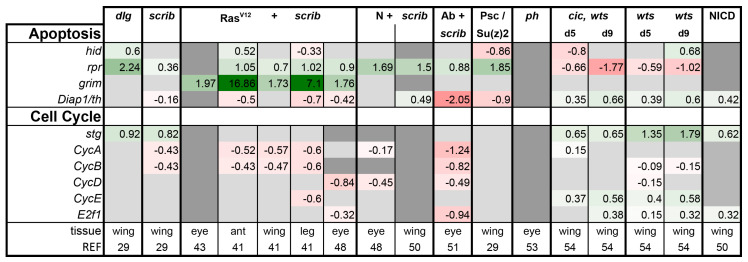
Expression of genes involved in apoptosis and cell cycle across tumors. Gene expression is displayed in log2FC. Color coding corresponds to a heatmap with a scale from −5 (dark red) to 5 (dark green). Light grey cells represent non-significant changes with *p* ≥ 0.05. Dark grey: the gene was not found in the datasets. Abbreviations: head involution defective, hid; repear, rpr; thread, th; string, stg; Cyclin A, CycA; E2F transcription factor 1, E2f1.

**Figure 10 ijms-21-04580-f010:**
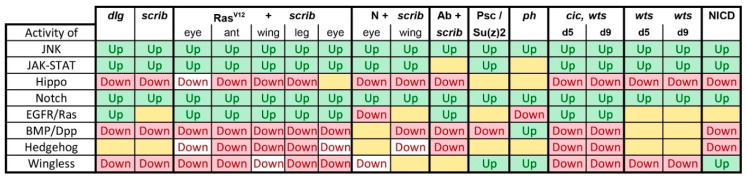
Overall summary of pathway activity across tumors. Yellow boxes indicate that the data were either inconclusive or there was no significant change in pathway activity based on target gene expression levels. White background indicates low confidence as the conclusion was based on a single data point.

**Table 1 ijms-21-04580-t001:** Common pathway members and their activity read-outs. A brief introduction of the key players of the pathways that were examined in this work. EGFR, Epidermal Growth Factor Receptor; Dpp, Decapentaplegic; Hh, Hedgehog; Wg, Wingless; JNK, Jun N-terminal Kinase; Jak/Stat, Janus Kinase/Signal Transducer and Activator of Transcription; Spi, Spitz; Vn, Vein; MAPK, Mitogen-Activated Protein Kinase; Cic, Capicua; Pnt, Pointed; Aop, Anterior Open; ERK, Extracellular Signal-Regulated Kinase; D, Delta; Ser, Serrate; Su(H), Suppressor of Hairless; Mam, Mastermind; E(spl), Enhancer of split; NRE, Notch Response Element; NICD, Notch Intracellular Domain; Tkv, Thickveins; Put, Punt; Mad, Mothers against Dpp; Med, Medea; Brk, Brinker; Shn, Schnurri; Dad, Daughters against Dpp; Sal, Spalt; Omb, Optomotor Blind; Ptc, Patched; Smo; Smoothened; Ci, Cubitus interruptus; Arr, Arrow; Dsh, Dishevelled; Sgg, Shaggy; Arm, Armadillo; βcat, Beta-catenin; dTCF, Drosophila T-cell Factor/Pangolin; Nkd, Naked; Egr, Eiger; Grnd, Grindelwald; Wgn, Wengen; Hep, Hemipterous; Bsk, Basket; Kay, Kayak; Jra, Jun-related antigen; Atf3, Activating transcription factor 3; Irbp18, Inverted repeat binding protein 18 kDa; CEBPG, CCAAT Enhancer Binding Protein Gamma; Pdp1, PAR-domain protein 1; puc, puckered; Mmp1; Matrix Metalloproteinase 1; TRE, Transcription Response Element; Upds, Unpaireds; Dome, Domeless; Hop, Hopscotch; Ds, Dachsous; Crb, Crumbs; Wts, Warts; Yki, Yorkie; Sd, Scalloped; Ex, Expanded; Diap1, Drosophila inhibitor of apoptosis protein 1; Ban, Bantam.

Pathway	EGFR/Ras	Notch	Dpp	Hh	Wg	JNK	Jak/Stat	Hippo
Ligand	Spi, Vn	D, Ser	Dpp	Hh	Wg	Egr	Upds	Ds
Receptor	EGFR	N	Tkv, Put	Ptc, Smo	Arr, Fz	Grnd, Wgn	Dome	Fat, Crb
Key Players	Ras, Raf, MAPK				Dsh, Sgg, Axin, APC	Hep, Bsk/JNK	Hop (Jak)	Hpo, Wts
Nuclear Factors	Cic, Pnt, Yan/Aop	N, Su(H), Mam	Mad, Med, Brk, Shn	Ci	Arm/βcat, dTCF	Kay, Jra, Atf3, Irbp18/ CEBPG, Pdp1	Stat-92E	Yki Sd
Activity Assays	anti-dp-ERK anti-Cic pnt-lacZ aos-lacZ cic-GFP	E(spl)m8-lacZ NRE-EGFP Su(H)-lacZ anti-NICD	dad-GFP brk-GFP anti-P-Mad anti-Sal anti-Omb anti-Brk	anti-Ptc dpp-lacZ ptc-lacZ	anti-Cut nkd-lacZ arr-lacZ	puc-lacZ anti-Mmp1 anti-P-JNK TRE-dsRed	10X STAT-GFP anti-P-STAT	ex-lacZ diap1-lacZ ban-GFP anti-Ex Yki::GFP

The color coding indicates the positive (green) or negative (red) effect of the protein on tissue growth if known.

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
