# Peer review of "Rounding up the Usual Suspects: Assessing Yorkie, AP-1, and Stat Coactivation in Tumorigenesis"

_ijms, 2020, doi:10.3390/ijms21134580_

Round 1

Reviewer 1 Report

Fisun Hamaratoglu et al overviewed previous reports of the several common pathways such as JNK, JAK-STAT, Notch, EGFR/RAS, Dh, and Hippo in the tumorigenesis of Drosophila tumor models. They concluded that JNK, JAK-STAT, Hippo, and Notch contribute to tumor development in all Drosophila cancer models tested. However, the manuscript focused on the Drosophila tumors, what is the significance of this manuscript? Since it is always known for these signaling pathways in human cancers and a huge amount of papers already established the roles of that these oncogenic signals in the different aspects of human tumor development, from tumor initiation to tumor growth, proliferation, and metastasis. Did the authors provide coactivation pathways that could be used for combinational drug cancer therapy? The manuscript should provide a more specific contribution/goal of this work. Minor comments:

1. TABLE 1, there is no green or red in table 1 as the table legendary indicated.

Reviewer 2 Report

The manuscript entitled “Rounding up the usual suspects: Assessing Yorkie, AP-1 and Stat coactivation in tumorigenesis” by Hamaratoglu and Atkins, presents the mined analysis of 8 pathways, reported transcriptomes of Drosophila imaginal discs tumors. Interestingly, the meta-analysis reported in this work suggests that in most of the cases the Hippo pathway is inactivated, but that the STAT, Jun Kinase and Notch pathways are over-activated. Some of this information was already known from different reports, however in this work, this information is integrated and new players of the different pathways were incorporated, in particular the role of Notch in tumor development in the fly.

I think that this is an interesting work that contributes to our understanding of cancer biology using Drosophila as model as well as it may be a reference article for researchers using Drosophila as a model organism to study cancer. However, before acceptance for publication, there are few concerns that need to be attended

1.-Most of the analysis consists in the generation of clusters derived from different transcriptomic experiments that are public. In these results, the expression differences are reported as Log2FC. The difference in gene expression calculated by this method, in general, consider that a significant fold change occurs when it is > to 1 or < to -1. In many transcripts reported in the different figures along with the manuscript, values are below 1 or greater than –1 and are indicated as a significant fold change.  An example is the EcR in figure 2. This point has to be clarified.

2.-The levels of overexpression of Myc in the Hippo pathway (figure 4) and in figure 8, in general, are lower than expected since in mammals, in most of the solid tumors, this oncogene is overexpressed. However, based on the data reported in this work, it seems not to be the case. It will be important to discuss a little bit more about this point.

3.-Figure 9 shows the expression of pro-apoptosis genes, and for the authors is a surprise that in many tumors there is an increase in the expression of these genes.  This is not a surprise since during the cell over-proliferation during the formation of the tumors, the cells are in stress, and even that there is an increase in proliferation, an important number of cells die via apoptosis. This has to be considered in this interpretation.

4.-The authors indicate that a color code in table 1, the key pathways are indicated, but there is no such color code in table 1.

Round 2

Reviewer 1 Report

 the concerns are addressed.